# The Detection of Vaccine Virus and Protection of a Modified Live, Intranasal, Trivalent Vaccine in Neonatal, Colostrum-Fed Calves with an Experimental Bovine Respiratory Syncytial Virus Challenge

**DOI:** 10.3390/pathogens13060517

**Published:** 2024-06-19

**Authors:** Stephanie Perkins-Oines, Nirosh D. Senevirathne, Greta M. Krafsur, Karim Abdelsalam, David Renter, Brent Meyer, Christopher C. L. Chase

**Affiliations:** 1Clinvet-South Dakota, 801 32nd Ave, Brookings, SD 57006, USAgreta.krafsur@cuanschutz.edu (G.M.K.); karim.abdelsalam@4rtilab.com (K.A.); 2Department of Diagnostic Medicine/Pathobiology, College of Veterinary Medicine, Kansas State University, Manhattan, KS 66506, USA; drenter@vet.k-state.edu; 3Beef Technical Services, Merck Animal Health, DeSoto, KS 66018, USA; brent.meyer@merck.com; 4Department of Veterinary and Biomedical Sciences, College of Agriculture, Food and Environmental Sciences, South Dakota State University, Brookings, SD 57007, USA

**Keywords:** bovine respiratory syncytial virus, bovine herpesvirus, vaccine virus detection, neonatal calves, maternal antibody, intranasal trivalent vaccine, virulent challenge, pulmonary lesions, serum neutralization antibodies

## Abstract

The efficacy of an intranasal (IN) bovine respiratory syncytial virus (BRSV) vaccine administered in the presence of passive immunity was assessed. Pooled colostrum was administered by intubation to 50 beef-dairy crossbred calves the day they were born. The calves were transported to a research facility and were blocked by age and sex, and randomly assigned into two groups: sham-vaccinated intranasally with a placebo (sterile water) or vaccinated with a trivalent (BRSV, bovine herpesvirus 1 and bovine parainfluenza 3) modified live viral (MLV) vaccine. The calves were 9 ± 2 days old when vaccinated (day 0). The calves were challenged by aerosolized BRSV on days 80 and 81 as a respiratory challenge. The study was terminated on day 88. Lung lesion scores (LLS) were significantly lower for calves vaccinated with trivalent MLV vaccine than those for calves that were sham-vaccinated. Serum neutralization (SN) antibody against BRSV in calves vaccinated with the trivalent MLV vaccine demonstrated an anamnestic response on day 88. After challenge, the calves sham-vaccinated with the placebo lost weight, while those vaccinated with the trivalent MLV vaccine gained weight. In this study, colostrum-derived antibodies did not interfere with the immune response or protection provided by one dose of the trivalent MLV vaccine.

## 1. Introduction

Bovine respiratory disease (BRD) is a significant health concern for beef and dairy cattle in the United States and globally, leading to substantial morbidity and mortality rates [1,2,3,4]. The financial impact of BRD on the US cattle industry is estimated to be around USD 800–900 million annually, primarily due to reduced weaning weights, lower average daily gains (ADGs), increased respiratory treatment costs, and higher feeding expenses [5]. This complex disease in cattle is caused by multiple factors, including stress, viral, and bacterial infections. Bovine respiratory syncytial virus (BRSV) is a common viral trigger for BRD, making the management of BRSV infections a crucial intervention in BRD control [6].

BRSV is a single-stranded enveloped negative sense RNA virus in the paramyxovirus family [6]. Clinical signs of BRSV typically peak 4 to 6 days post-infection. Infected calves usually develop a fever 2–6 days post-exposure, with body temperatures reaching up to 104 °F. In addition to fever, the animals may show lethargy, reduced appetite, increased respiratory rate, labored breathing, coughing, mucopurulent nasal discharge, and wheezing lung sounds [6].

One widely recognized strategy for disease prevention is the transfer of maternal antibodies to newborn calves. The transfer of colostral antibodies from the mother plays a critical role in protecting newborn animals against diseases [7,8]. However, maternal antibodies can pose a challenge as they may interfere with vaccination, another commonly used disease management method [7,8,9]. Nevertheless, considering variations in herd health conditions and individual producer preferences, a calf vaccination program is essential to effectively complement colostrum administration in the United States. Maternal antibodies shield the calf from potential pathogens but may also interfere with antigens present in the vaccine, hindering the calf’s ability to mount an effective immune response [9,10,11]. To overcome interference from maternal antibodies, producers typically administer vaccinations to calves once the antibody concentrations have decreased sufficiently [9,11]. For successful vaccination against BRSV, maternally derived antibody titers should be below 1:4 [8,12,13]. The decline in maternal antibodies can create vulnerability to BRSV within the herd, making it challenging to establish standardized calf vaccination schedules.

While many vaccines are approved for use in young calves under 30 days old, most studies for licensing purposes do not assess the impact of maternal antibodies (often referred to as in the face of maternal antibody, IFOMA) [9,14,15]. Additionally, the immune system maturation with age means some young calves under 30 days old may exhibit immunological immaturity, complicating vaccination in the presence of maternal antibodies. The majority of studies on BRD vaccines have concentrated on administering vaccinations around the time of weaning (before, during, or after) and subsequently monitoring their effects on BRD morbidity and mortality. Another issue frequently encountered with IN viral vaccines is the use of molecular diagnostics in young calves to detect viral respiratory agents. The use of IN viral vaccines has been fraught with two “diagnostic issues”—the detection of vaccine virus resulting in a misdiagnosis that the animals were “infected” or, conversely, the “lack of detection” of vaccine virus in IN-vaccinated calves leading to the “conclusion” that the calves were not vaccinated. There have been few published studies measuring virus levels following IN vaccination in young calves. The aim of this study was two-fold: (1) to assess the protection of immunity following vaccination at 9 ± 2 days of age in the presence of maternal antibodies, employing a BRSV challenge model approximately 80 days post-vaccination and (2) to measure the kinetics of the vaccine virus in vivo following IN vaccination.

## 2. Materials and Methods

Personnel who monitored the calves, obtained samples, scored the pulmonary lesions, or performed laboratory procedures were not aware of which treatment group a calf was assigned.

Animals, housing, and feeding: Fifty beef-dairy crossbred male and female calves from a single source were acquired for this study. At birth, calves were separated from their respective dam before suckling. Blood was obtained for serologic determination of exposure to BRSV. An ear notch biopsy using an enzyme-linked immunosorbent assay (ELISA) was obtained to determine persistent infection with bovine virus diarrhea virus (BVDV) using IDEXX Antigen Capture kit (IDEXX, Portland ME, USA). Pooled colostrum, characterized for antibody titers to BHV-1 (1:128) and to BRSV (1:128), was then administered by intubation and 6 h after the first feeding. Within a few hours of birth, the calves were examined and transported to the contract research facility and, within 24 h, received 2 feedings of pooled colostrum (4 L/feeding). They were identified with individual numbers on ear tags and housed in individual hutches. Calves were observed daily by trained personnel. The animals were weighed at vaccination, 0 days post-vaccination (DPV) (9 ± 2 days of age), 14 DPV, 80 DPV (days post-challenge (DPC) 0) and 88 DPV (DPC 8). Signs of illness or injury were recorded, and care was provided and documented according to protocols.

Fresh water was available ad libitum to each calf throughout the study. The calves were fed milk replacer twice daily and grain supplement was offered from 3 days of age. When the youngest calf was 56 days old, all calves were weaned as a single group. They consumed 2.5 to 3 lbs. of pelleted feed per head, which met nutritional requirements. The animals were weaned using a step-down procedure over 8 days. At the end of the 8-day weaning period, calves were fed only the pelleted feed. Calves were then commingled and had access ad libitum to hay in addition to the grain supplement. Calves that died or were euthanized were submitted to the Animal Disease and Diagnostic Laboratory (ADRDL) at South Dakota State University for diagnostic necropsy.

Vaccine and Experimental treatments: Calves were stratified by sex and age to ensure even distribution among the two groups and then randomly assigned to the treatment group. The vaccine was administered to calves when they were 9 ± 2 days old (Figure 1).

The trivalent vaccine (Bovilis Nasalgen 3-bovine herpesvirus 1 (BHV-1), bovine respiratory syncytial virus (BRSV), and bovine parainfluenza 3 (PI3)) was reconstituted, handled, and administered according to label directions. There were 50 animals, 25 in each of the two treatment groups. On day 0, calves in the vaccine group (VG) were vaccinated intranasally (IN) with one dose of the trivalent vaccine (2 mL) instilled in a single nostril and one dose (2 mL) of a placebo (sterile water) administered similarly to calves in the placebo group (PG). The mean BRSV SN serum antibody titer was 6.29 log2 for the trivalent vaccinated group and 6.16 log2 for the sham-vaccinated placebo group at the time of vaccination. The mean BHV-1 SN serum antibody titer was 5.55 log2 for the trivalent vaccinated group and 5.64 log2 for the sham-vaccinated placebo group at the time of vaccination. On day 0, there was no significant difference (*p* = 0.22) in body weight for calves in VG (mean = 93.36 lb., SEM = 2.68), and for calves in PG mean = 98.04 lb., SEM = 2.68). There was also no significant difference (*p* = 0.41) of total protein in serum for calves in VG (mean = 6.93 g/100 mL, SEM = 0.09) and for calves in PG (mean = 6.83 g/100 mL, SEM = 0.09). BRSV and BHV-1 serum neutralization (SN) titers for each calf were <2 prior to administration of the pooled colostrum. None of the calves was persistently infected with BVDV.

Calves were observed daily by trained personnel for clinical signs of illness or injury, which were recorded and treated according to protocols. After enrollment, two (2) calves (one from each treatment group) died or were euthanized before challenge.

### 2.1. Sample Collection

Blood samples: Blood samples for serum analysis were obtained from the calves via jugular venipuncture at 7–21 day intervals for approximately 10 weeks prior to challenge, the day before challenge (−1 DPC), and at necropsy (8DPC) (Figure 1). Briefly, the animal was restrained, and 12.5 mL of blood was collected from the jugular vein. Blood samples were used to measure total protein at 24 h to determine colostral absorption and for serum neutralization analysis to determine antibody levels against BRSV and BHV-1.

Nasal Swabs: Nasal swabs were collected from all calves on days 0, 1, 3, 5, 7, 9, and 14 DPV for both BRSV and BHV-1 for polymerase chain reaction (PCR) assays and 0, 1, 5, 9 and 14 DPV for virus isolation (Figure 1). Nasal swabs were collected from all calves on days −1, 0, 3, 5, 7, and 8 DPC for BRSV for VI and PCR assays. To collect samples, calves were initially manually restrained in the hutches. Following comingling, the calves were physically restrained by head catches. Once their head was gently restrained, a sterile swab (Copan) was inserted gently into one naris, rotated briefly, then removed and deposited into a sterile conical tube containing 2 mL of transport medium (Dulbecco’s Minimum Essential Media, Eagle containing 2× antibiotics/antimycotics and 1× L-glutamine) and transported to the laboratory for processing on at 4 °C.

### 2.2. Sample Processing and Analysis

Cells and virus: Bovine turbinate cells were harvested from BVDV-negative calves using established methods. Primary cultures were passaged 2–3 times, trypsinized, counted, and aliquoted in 1 mL aliquots, and then frozen at −196 °C in liquid nitrogen for subsequent use. Madin–Darby Bovine Kidney (MDBK) cells were obtained from the American Type Culture Collection, Manassas, VA, USA. The G-01 strain of BRSV, a kind gift from Amelia Woolums, Mississippi State University, was used for challenge and serum neutralization assays [15]. Cells were plated at 5 × 10^4^/mL. The Colorado strain of BHV-1 was obtained from the National Veterinary Services Laboratory, Ames, IA, USA, and was used for BHV-1 serum neutralization. Cells were cultured with 5% FBS with MEM at 37 °C.

Serum neutralization assay: Serum was harvested from whole blood as follows: samples were centrifuged at 1500× *g* for 15 min at 4 °C. Serum was decanted and tested for the presence of neutralizing antibody against BRSV and BHV-1. Two-fold serial dilutions of sera (1:2 to 1:256) in duplicate were made in 96-well microtiter plates. A constant amount of virus (<500 tissue culture infectious dose-50% (TCID_50_)) was added to each serum dilution well of the plate as appropriate. After incubation of the serum/virus mixtures for 2 h, 200 μL of the mixture was used to inoculate bovine turbinate cell (BT) monolayers contained in 96 well microtiter tissue culture plates. Plates were incubated at approximately 37 °C with 5% CO_2_ for 3 to 8 days (3 days for BHV-1 and 8 days for BRSV) before being evaluated for virus-induced cytopathic effect (CPE) [16,17]. The positive sera control for BRSV or BHV-1 was obtained from the National Veterinary Services Laboratory, Ames, IA. The serum titer was determined to be the last dilution that inhibited CPE and was presented as a reciprocal value. Geometric mean values were then calculated using log2 titers.

Nasal virus shedding assay: Nasal swabs in transport media were processed (vortexed and centrifuged at 600× *g* for 5 min at 4 °C) and aliquoted for BRSV or BHV-1 VI or BRSV or BHV-1 PCR assays. For BRSV VI, dilutions using the transport media of each sample were made, and 200 μL was added in triplicate to BVDV-free bovine turbinate (BT) cell monolayers in microtiter tissue culture plates [16]. Plates were incubated for 9 days without a media change at 37 °C in 5% CO_2_. Positive control for BRSV was the G-01 strain, and mock-infected wells were the negative control. Plates were stained with BRSV monoclonal antibody (GeneTex, Irvine, CA, USA) and read using a fluorescence microscope (Olympus CK400 fluorescence microscope, Olympus, Melville, NY, USA) [17]. Results were considered positive if BRSV intracellular staining was seen in inoculated wells after one blind passage. For BHV-1 VI, serial 10-fold dilutions of samples using transport media were made, and 200 μL of diluted sample was added in triplicate to MDBK cell monolayers in microtiter tissue culture plates. Samples on culture plates were incubated for 5 days at 37 °C with 5% CO_2_ before being evaluated by cytopathic effect and BHV-1 immunofluorescence staining. The positive control for BHV-1 was the Colorado strain, and mock-infected wells were the negative control. Samples were considered positive for the BHV-1 virus if the cytopathic effect or virus-specific fluorescence was observed in inoculated cells after one blind passage [18,19]. TCID_50_ was calculated for BRSV or BHV-1 using a Spearman–Karber method [20]. BRSV and BHV-1 PCR nasal sample aliquots and lung were submitted to the Molecular Diagnostics Section at the Animal Disease and Research Diagnostic Laboratory (ADRDL) at South Dakota State University Brookings SD for BHV-1 [21] analysis using PCR and BRSV [18] analysis using real-time Reverse Transcription Polymerase Chain Reaction (RT–PCR). The same positive and negative controls were used for PCR as for VI. An aliquot of the titrated BRSV G-01 virus was diluted in ten-fold dilutions (1:10–1:1,000,000) and assayed by real-time RT–PCR to develop a standard curve to correlate cycle threshold (CT) values to TCID_50_/mL in a semiquantitative manner [22].

Challenge and Clinical Signs: The challenge, clinical, and pathological examination were previously described [16]. The BRSV challenge material was passed through colostrum-deprived, unvaccinated calves that had been raised in a controlled, isolated environment prior to the challenge. The challenge material was generated by infecting 2 calves with BRSV for two consecutive days by aerosolization of 4 mL of the G-01 strain of BRSV. The calves were euthanized at 8 days post-challenge (DPC), and the bronchoalveolar fluid was collected, filtered, and titered for BRSV. Prior to use, the challenge material was also tested for the presence of BVDV by RT–PCR [23] and cultured on MacConkey, blood agar, and Sabouraud plates (Thermo–Fisher, Waltham MA, USA) to detect bacteria and molds. Virulent BRSV was administered (24 calves per treatment group) on two consecutive days (day 80 and day 81) by aerosolization (Figure 1). The volume administered (4 mL) had an average titer of 1.25 × 10^5^ virus TCID_50_/mL using a nebulizer and mask to aerosolize the challenge material for intranasal inhalation by the calves (DriveTM Reusable Nebulizer Kit, Port Washington, NY, USA). Clinical disease parameters, including attitude, body temperature, and general respiratory signs, were monitored for 8 days following challenge by trained personnel blinded to treatment groups (Figure 1). Calves were considered to be pyrexic when body temperatures were 103.5 °F or greater. Each calf was visually examined and scored for signs of abnormal respiration, nasal and ocular discharge, and depression, using a scale of 0–3, except for nasal discharge, which was on a scale of 0–4. Briefly, an abnormal respiration score was given if an animal had short/rapid breathing (1), mild dyspnea (2) or severe dyspnea (3). Coughing was noted as present (1) or absent (0). Nasal discharge scores ranged from no discharge (0), mucous (1), mucopurulent (2), purulent (3) or blood-tinged (4). Ocular discharge scores ranged from normal slight serous (0), moderate serous (1), moderate mucopurulent (2) or purulent (3). Attitude (Depression) was scored no depression (0), mild (1), moderate (2), or severe depression (3). One animal in the placebo group died on day 87, 7 days following challenge with respiratory distress.

Necropsy: The 2 calves that died before challenge (one from each group) and the placebo calf that died at day 87 were necropsied. All calves remaining in the study (47 calves—24 vaccinates and 23 placebos) were euthanized by barbiturate overdose on day 88 (8 days post-challenge) (Figure 1). Lungs with trachea attached were removed from each calf. Lesions of the lungs were scored as a percentage of the lung involved. Each lung lobe was examined in its entirety by a board-certified pathologist, and the extent of lung involvement was estimated as a percentage of each lobe. Once the lung lesion scores were collected and recorded, their percentages were entered into an Excel spreadsheet, and the results were adjusted to reflect the proportion of the lesion relative to the whole. Specifically, each lung lesion score was multiplied by the percent of the total lung area for each lobe of the bovine lung to determine what proportion of the lungs were damaged by the challenge. A score for both lungs was calculated based on the estimation of the percent of lesions [24] in each lobe as follows: (Left Cranial × 0.05) + (Left Middle × 0.06) + (Left Caudal × 0.32) + (Right Cranial × 0.06) + (Right Posterior Cranial × 0.05) + (Right Middle × 0.07) + (Right Caudal × 0.35) + (Accessory × 0.04) = Total Lung Score (TLS). Sterile forceps, scissors, and disposable scalpels were used to extract pulmonary tissue sections for BRSV RT–PCR, BRSV virus isolation (VI), routine histopathology, and BRSV immunohistochemistry (IHC). Two representative lung samples were collected from each calf for virus isolation (VI) and polymerase chain reaction (PCR) assays. Tissues for viral PCR testing and VI were placed in sterile Whirl Pak^®^ bags (one for PCR and one for VI) and labeled with the study number, date of sample collection, study day, calf ID, and lobe ID. A minimum of two pulmonary tissue sections were collected from characteristic BRSV pulmonary lesions for routine histopathology and IHC. Tissue sections for H&E and IHC staining were fixed in 10% neutral buffered formalin (NBF). Formalin containers were labeled with the study number, date of sample collection, study day, calf ID, and lobe ID. Formalin-fixed samples were transferred from the study site to the ClinVet-SD laboratory at ambient temperature. Fresh samples of the lung lesions were transported on ice packs to the laboratory. Formalin-fixed tissue sections from characteristic BRSV lesions were trimmed by the investigation pathologist (GMK) and placed in tissue cassettes. The tissue cassettes were transferred to the ADRDL, where the tissues were embedded in paraffin wax, sectioned at 3–5μm, and stained with hematoxylin and eosin (H&E) and by IHC to detect BRSV antigen utilizing protocols employed by histotechnologists at the SDSU Animal Disease Research and Diagnostic Laboratory. For BRSV IHC, appropriate positive and negative control pulmonary tissue sections were included in each run. The investigation pathologist (GMK) analyzed, interpreted, and documented all slides by means of light microscopy without prior knowledge of the animal’s experimental treatment. Interpretation of H&E sections included the extent of consolidation expressed as a percentage of the tissue sections and the presence of BRSV-related pathologic alterations, including bronchointerstitial pneumonia, necrotizing bronchiolitis, viral syncytia, exudative and proliferative alveolitis, emphysema and bronchus-associated lymphoid tissue (BALT) hyperplasia. The extent and localization of BRSV antigen in the airway epithelium, exfoliated cellular detritus, and alveolar histocytes were evaluated in IHC tissue sections.

Statistical analysis: Data were analyzed using the animals as independent experimental units.

General (normal distribution) and generalized (non-normal distributions) linear (mixed) models were used for continuous and non-continuous outcomes, respectively, with PROC Glimmix (SAS version 9.4, Cary, NC, USA). Data that were not normally distributed were transformed prior to analysis. Repeated measures analyses included fixed effects for treatment, day, and interaction between treatment and day, as well as random effects to account for covariances among observations at different sampling times within animals. Pair-wise comparisons were adjusted for multiple comparisons using Tukey methods. Model-adjusted means and corresponding standard error of the mean (SEM) are reported, and *p* < 0.05 was considered significant.

## 3. Results

### 3.1. Nasal Shedding of BHV-1 or BRSV Virus Following Vaccination

BHV-1: Calves vaccinated with the trivalent MLV vaccine had detectable BHV-1 virus beginning at day 1 post-vaccination, as detected by PCR in 8% of the calves (2 of 25; Figure 2). At day 3, 40% of the animals (10 of 25) were PCR positive.

This increased to a maximum of 64% (16 of 25) at day 5. The number of positive animals began decreasing at day 7 to 60% (15 of 25), 48% (12 of 25) at day 9, and all the animals were negative at day 14. PCR cycle threshold (CT) values (the smaller the CT value, the greater the amount of virus) had a similar pattern to the number of BHV-1 positive animals with the highest levels of virus at day 5 and the animals negative at day 14. Seventeen (17) of the 25 calves shed BHV-1 at least once during the 9 days following IN vaccination. There were 55 total BHV-1 PCR detections, and 15 of the 17 BHV-1 positive animals shed BHV-1 for more than 3 days. Average CT values decreased from day 1 to day 5 (day 1, 29.6; day 3, 26.8; and day 5, 24.0). Average CT values increased from day 5 to day 14 (day 5, 24.0; day 7, 24.5; day 9, 26.3 and day 14 >40). BHV-1 virus isolation (VI) results paralleled the PCR results both in peak percentage of BHV-1 positive animals and in the virus levels on those days that both assays were run. At day 1, 12% (3 of 25) of animals were VI-positive with an average titer of 2.07 log_10_ TCID_50_/mL, peaking at day 5 at 56% (14 of 25) and average titer of 4.57 log_10_ TCID_50_/mL. VI-positive animals decreased at day 9 to 40% (10 of 25) with an average titer of 3.02 log_10_ TCID_50_/mL. In comparing the two virus detection methods, ~90% of the positive animals were positive with both methods. There was a single control animal BHV-1 positive on a single day from all the control animals collected for the 14 days following vaccination (Figure 2). The animal was BHV-1 PCR positive only on day 5 and negative for PCR and VI for the other collection days.

BRSV: Unlike BHV-1, few animals shed BRSV following vaccination (Figure 3).

At day 1, 4% of the calves (1 of 25) were BRSV PCR positive. At day 3, 8% of the animals (2 of 25) were PCR positive, and 16% (4 of 25) at day 5. The number of positive animals began decreasing at day 7 to 12% (3 of 25) and 8% at day 9 (2 of 25). Seven (7) of the 25 animals were BRSV PCR positive. There were 12 BRSV detections, and 7 of the BRSV detections were from 2 animals (3 and 4 positive detections). The remaining 5 animals had a single day of BRSV detection. The CT values were 29.8 or greater at all time points with no changes in calculated semiquantitative titers over time. BRSV VI detections were also very low, with 3 positive BRSV VI detected on day 1 with titers less than 2 log_10_ TCID_50_/mL. Calves that were sham-vaccinated with a placebo were negative for BRSV throughout the post-vaccination period (Figure 3).

### 3.2. Clinical Signs

Febrile response: There was a significant (*p* < 0.01) interaction between treatment and day, as well as the main effect of treatment (*p* < 0.01) and of day (*p* < 0.01) on rectal temperature. Body temperatures of the two groups were similar for the first 4 days post-challenge (DPC) (days 81–84) (Figure 4).

Mean rectal temperatures in the placebo group on DPC 5–7 (days 85–87) were ~1 ˚F higher than the vaccinates (Figure 4). DPC 5–8 (days 85–88) were significantly (*p* < 0.01) higher for sham-vaccinated placebo calves as compared to the vaccinated calves with the trivalent MLV vaccine (Figure 4).

Respiratory signs and cough: Overall effects of Treatment were significant (*p* < 0.01) for mean respiratory scores after challenge (Figure 5).

The mean respiratory scores over the 8-day post-BRSV challenge period were 4× greater in the placebo animals as compared to vaccinates (Figure 5). The overall probability of coughing (measured as a binomial: present or absent) was 1.8× times higher. Cough was significantly (*p* = 0.01) higher in the placebo group.

Mortality: After enrollment and before the BRSV challenge, one calf in each of the vaccinates and placebo groups was euthanized for animal welfare consideration due to an unresponsive joint infection that resulted in severe lameness. Seven days (day 87) after challenge, one calf sham-vaccinated with the placebo died (Figure 5). On necropsy of the placebo calf, there was severe lung involvement consistent with BRSV that was confirmed by virus isolation and PCR. No bacteria associated with bovine respiratory disease were cultured.

Weight gain following vaccination and challenge: Vaccination had no significant effect on weight gain in the first 21 days following vaccination (Figure 6).

During the post-challenge time, calves vaccinated with the trivalent MLV vaccine gained 8.25 lb. (SEM = 1.98) body weight while those sham-vaccinated with the placebo lost weight (−5.39 lb.; SEM = 2.02) (*p* < 0.01) (Figure 6). There was no overall body weight effect over the 88-day period (Figure 6).

Serum neutralization antibodies against BRSV and BHV-1: Titers of BRSV SN antibodies for all calves were <2 at the time of enrollment (Figure 7).

The mean BRSV antibody titers decreased after enrollment until challenge (Figure 7). After vaccination, there was no significant (*p* = 0.55) effect of treatment. After challenge, there was a significant (*p* < 0.01) effect of vaccination as on day 88, the mean BRSV SN titer was significantly (*p* < 0.05) higher for calves vaccinated with the trivalent MLV vaccine than for calves sham-vaccinated with placebo (Figure 7). Fourteen (14) of the 25 vaccinated calves had SN antibody increases from DPC 5 (day 85) to DPC 8 (day 88), 8 stayed the same, and 2 decreased. In contrast, in the sham-vaccinated placebos, BRSV antibody levels exhibited classic maternal antibody decay, with 0 of the 23 animals having an increase, 12 of 23 staying the same, and 11 decreasing. Interestingly, 3 of the trivalent vaccinated group BRSV titers were <2 (seronegative) at DPC 5, and they had BRSV SN titers ranging from 2 to 64 three days later on DPC 8, demonstrating memory and anamnestic response IFOMA. For BHV-1, BHV-1 SN titers were measured over the 88-day period, and the antibodies decreased at the same level between the two treatments, and there was no significant effect (*p* = 0.15) of vaccination on SN antibodies against BHV-1.

Shedding of BRSV Post-Challenge: The sham-vaccinated placebo calves did not shed BRSV until after challenge with virulent BRSV on day 80. All (100%: 24 of 24 DPC 3–6; 23 of 23 DPC 7) of the placebo calves were positive by RT–PCR BRSV in nasal secretions from DPC 3, 5 and 7 (days 83, 85 and 87) (Figure 3). On DPC 8, 97% (22 of 23) sham-vaccinated placebo calves were positive by RT–PCR (Figure 3). A higher percentage of the vaccinated animals were positive by RT–PCR BRSV in nasal secretions from DPC 3, 5, and 7 (days 83, 85, and 87; 92%, 96%, and 88%, respectively). On DPC 8 (day 88), the number of tri-vaccinated animals positive for RT–PCR BRSV dramatically decreased to 46% (11 of 24) (Figure 3).

Levels of BRSV detected by RT–PCR or virus isolation also differed between the sham-vaccinated placebo and the trivalent vaccinated groups throughout the challenge period (Figure 8 and Figure 9).

The levels of BRSV in nasal secretions, as measured by semiquantitative RT–PCR, were higher at all time points in the placebo group (Figure 8), peaking at DPC 5 (day 85), and was 1–2 log_10_ significantly higher on DPC 5, 7 and 8 (*p* < 0.01). Similar results were seen with the BRSV infectious virus shed in nasal secretions. The virus was isolated at all time points post-challenge in the placebo group (Figure 9). The placebo group peaked at DPC 5 (day 85) with almost 2 log_10_ infectious BRSV being shed and was 0.5–1 log_10_ significantly higher on DPC 5 and 7 (*p* < 0.01). In contrast, the trivalent vaccinated group shed less than a l log_10_ of infectious virus and was negative on DPC 8 (day 88) (Figure 9).

### 3.3. BRSV Lung Evaluation

BRSV Virus Detection: The lung samples were tested for BRSV following necropsy. Ninety-two percent (22 of 24) of the sham-vaccinated placebo animals were positive as compared to 71% (17 of 24) of the vaccinated animals. Like the nasal samples, levels of BRSV as detected by RT–PCR were almost 2 log_10_ significantly lower in the trivalent vaccinates as compared to the placebo (Figure 10).

Interestingly, the sham-vaccinated placebo animal that died on DPC 7 had a CT value of 17.14, which would be 3 l log_10_ higher than the highest nasal sample at DPC 5. BRSV virus isolation was negative on all lung samples/

Gross Lung lesion scores: Pulmonary lesions were visualized in all experimental animals, irrespective of the treatment group. Lung lesions were reduced in the trivalent vaccinated animals (Figure 11 and Figure 12).

Mean LLS (SEM) (17.49% (2.33)) for calves vaccinated with the trivalent MLV vaccine were significantly (*p* < 0.01) lower than that (33.21% (2.99) for calves sham-vaccinated with the placebo (Figure 11). However, the distribution and magnitude of pulmonary pathology varied considerably between vaccinates and placebos (Figure 12A–D). Affected pulmonary parenchyma was pale red to plum red, consolidated due to atelectasis, and often surrounded or divided pale pink non-collapsing, hyperinflated lung (Figure 12B,D). Marked subpleural and interlobular edema fluid and emphysema accentuated individual lobules. Consolidated parenchyma was firm and cut crisp with a knife, while hyperinflated lobules were rubbery on the cut section and often exuded clear frothy edema fluid.

Microscopic Findings: Fulminant bronchointerstitial pneumonia was visualized in pulmonary tissue sections from 48/48 animals, and necrotizing bronchiolitis was detected in 15/48 animals (Figure 12E), with all but one of 15 belonging to the placebos. Caseonecrotic pneumonia superimposed upon bronchointerstitial pneumonia in pulmonary tissue sections from one placebo animal inferred mycoplasma co-infection owing to impaired lung defenses and permissive bacterial infection resulting from viral injury. Viral syncytial cells were observed in 2/48 animals, both placebos (Figure 12E). Varying degrees of BALT hyperplasia were visualized in 47/48 animals. Often, tissue sections depicted reparative efforts following lymphocyte-mediated lysis of virus-infected cells. These changes included hyperplastic bronchiolar epithelium, disappearance of syncytial cells with clearance of viral antigen, and alveolar tombstoning, i.e., cuboidal epithelialization of the alveoli often labeled Type II pneumocyte hypertrophy and hyperplasia.

Immunohistochemistry: BRSV immunosignaling was restricted to a limited number of animals (7/48), all of whom were placebos. Antigen expression was visualized in the airway epithelium, exfoliated cellular detritus occluding airway lumina, and alveolar histiocytes (Figure 12F). Limited immunolabeling of pulmonary tissues indicates clearance of viral antigen.

## 4. Discussion

Managing BRSV and the resulting secondary bacterial continues to be an enormous problem for beef and dairy cattle producers nationwide. The complications associated with disease prevention IFOMA and the economics of food production create a strong demand for better vaccination strategies and vaccines. The inhibition of BRSV vaccines administered both parenterally and intranasally by BRSV maternal antibodies has been well documented [11,25,26]. BRSV infection frequently occurs in young animals, and IFOMA vaccination needs to occur to minimize disease risk in production systems. The primary purpose of this study was to evaluate the efficacy of a trivalent MLV IN vaccine to stimulate the immune response and protect against virulent BRSV challenge with BRSV in the face of maternal antibodies (IFOMA) acquired from colostrum. This trivalent MLV IN vaccine reduced BRSV-associated disease in two studies in non-suckled, BRSV seronegative calves administered vaccine at ~5–7 days of age when challenged at 30 [27] or 78 [28] days post-vaccination. By controlling and defining the maternal transfer in the calves involved in the study, we hoped to evaluate the specific interactions between IN vaccination and passive acquired immune systems in young calves. To evaluate vaccine efficacy, maternal antibody transfer was standardized to provide a consistent baseline level to compare the vaccine group to the placebo group. The pooled colostrum utilized in the study contained BRSV titers of approximately 1:128, and the average antibody titer in the calves at 36 h was found to be half that (1:64), supporting the common paradigm that complete maternal antibody transference does not occur [25]. This level of antibody was similar to levels we had seen in a previous study [17] using a parenteral vaccine and in a BRSV intranasal vaccine study where calves had been fed undiluted colostrum [14]. In that study, calves from a large colostrum-fed cohort were screened, and 22 calves with similar BRSV maternal antibody titers were enrolled. The establishment and documentation of the calves’ BRSV antibody levels from birth were critical because the goal was to evaluate the animal’s response to MLV IN mucosal exposure when the maternal antibody was at low levels, thus elucidating the actual effect of an IN trivalent MLV vaccine administered to calves IFOMA. This is the second study where the maternal antibody was characterized and controlled to “normalize” the effect of vaccine response IFOMA on the BRSV viral challenge. While previous studies [12,29] have helped to define the half-life and decay of maternal BRSV antibody in colostrum and calves, the animals in this study were monitored, and the BRSV SN titer was (<1:8) at challenge. The clinical, virological, immunological, and pathological differences following BRSV between the trivalent IN-vaccinated calves and the placebo indicated that there is a distinct advantage to IN MLV vaccination IFOMA. Trivalent vaccinated calves had a reduction in the duration and severity of disease that was statistically significant and was similar to that seen with another IN BRSV vaccination IFOMA [14]. The decrease in the number of animals shedding virus (Figure 3), viral burden (Figure 8 and Figure 9), reduction of clinical signs (Figure 4 and Figure 5), and decreased lung viral burden and pathology (Figure 10, Figure 11 and Figure 12) indicate that the vaccinated calves mounted a protective response.

The administration of mucosal-based vaccines such as the trivalent IN MLV has been traditionally thought to result in primarily a mucosal response with little systemic response [30]. What was surprising was the significantly higher BRSV SN titer for calves vaccinated with the trivalent MLV vaccine that occurred on day 88, 8 days post-challenge, indicating an anamnestic response after “systemic priming” with a mucosal-delivered vaccination (Figure 7). The original publications that were done with Nasalgen IP, which contained the BHV-1 and bovine parainfluenza virus 3 (these are the same viruses in Nasalgen 3), demonstrated a BHV-1 SN response that was boosted following a BHV-1 challenge [31]. This systemic BRSV boost was not seen following the challenge in our previous study using an adjuvanted parenteral MLV BRSV vaccine [17]. The more active immune activation that occurs with the “normal” nonmutated BHV-1 in mucosal replication and immune interaction may provide an “enhancement” for the development of BRSV systemic immunity. To further support this hypothesis, a recent publication using the same BRSV and BHV-1 strains in the trivalent vaccine as monovalent vaccines demonstrated higher BRSV systemic antibody response in animals given a concurrent IN BRSV and IN BHV-1 than the animals that received a monovalent IN BRSV vaccine [10]. In contrast, animals vaccinated IN with a combination of a temperature-sensitive mutant BHV-1 vaccine and BRSV did not stimulate any BRSV SN response [30].

Another important finding was the effect of vaccination on BRSV shed. The level of infectious BRSV virus detected by PCR or shed as measured by VI was lower in trivalent vaccinated animals, and it was shed for less time. Additionally, the number of trivalent vaccinated animals shedding infectious BRSV virus was much less than the sham-vaccinated placebo calves. Similar to levels of BRSV detected via PCR or VI (Figure 8 and Figure 9), more sham-vaccinated placebo calves shed infectious BRSV in nasal secretions throughout the challenge period. Virus was isolated from 46% (11 of 24) on DPC 3, 92% (22 of 24) on DPC 5, 52% (12 of 23) on DPC 7, and 13% (3 of 23) on DPC 8 of the placebo calves. In the vaccinated animals, although there were a similar number of vaccinated animals shedding as placebos at DPC 3, 54% (13 of 24) vs. 46%, respectively, there was a large difference at DPC 5 were only 50% (12 of 24) of vaccinated calves were shedding infectious virus compared to 92% of placebo calves. Similarly, at DPC 7, only 13% of vaccinated calves (3 of 24) were shedding infectious BRSV compared to 52% of placebos. On DPC 8, no virus was shed in the vaccinates compared to 13% in the placebo calves. This decrease in the shedding of infectiousness will enhance herd immunity as less virus will be available to infect susceptible animals.

A second objective was to detect IN-administered vaccine viruses to provide insight into the duration of detection and levels of viral replication in nasal samples IFOMA, as nasal samples are frequently used for the diagnosis of calf respiratory disease. One of the areas of great confusion with IN nasal vaccines is two “diagnostic issues”—the detection of vaccine virus resulting in misdiagnosis or, conversely, “the lack of detection” of vaccine virus in IN-vaccinated calves (did they even receive the vaccine?). In this study, we found that BHV-1 from the trivalent IN MLV vaccine was detected by PCR as late as 9 days post-vaccination (Figure 2) from nasal swabs. PCR virus detection peaked at DPV 5 and was negative at DPV 14 (Figure 2). The virus was detected in a small percentage of animals at DPV 1. The highest level of BHV-1 was also detected at DPV 5. Virus isolation had a similar temporal pattern and was less sensitive than PCR (i.e., on DPC 5, 16 animals were detected by PCR vs. 14 with VI). On DPV 1, 3 of 25 (12%) vaccinated animals were BHV-1 VI-positive and peaked at 56% (14 of 25) at DPC 5. The number of positive BHV-1 VI declined to 40% (10 of 25) at DPV 9 and was negative at DPV 14. The highest viral titers were at DPC 5 when the average titer was 4.57 log_10_ TCID_50_/mL. These results agree with the original work done with Nasalgen IP (aka Bovilis Nasalgen IP) that demonstrated a similar temporal and peak VI detection [30]. The detection of BRSV following vaccination was much less, peaking at 16% at DPV 7 (Figure 3), and BRSV was only detected by PCR in 12 samples out of the 125 samples from trivalent IN-vaccinated animals. Alternately, BHV-1 was detected by PCR in 55 samples out of the 125 samples from trivalent IN-vaccinated animals. In another study, 30 beef calves were vaccinated intranasally within 24 h of birth with a BHV-1, BRSV, and PI3 vaccine (Inforce 3, Zoetis Animal Health) and sampled once between 1 and 7 days post-vaccination for BRSV [32]. Ten of these animals were revaccinated 14 days later and again sampled once between 1 and 7 days post-vaccination for BRSV. At both time points, all the animals were negative for BRSV using the same BRSV RT–PCR assay and laboratory used in this study [31]. This second study further supports the low level of detection of the IN-administered BRSV vaccine virus seen in our study.

To summarize, the results of this study support the immunogenic and protective findings of other studies with the trivalent MLV vaccine used in this study [27,28]. An important finding was that in the presence of colostral antibodies, animals vaccinated with an IN trivalent MLV vaccine were healthier than placebo calves when challenged 80 days after vaccination. Clinical, virological, immunological, and pathological findings all supported a significant advantage for vaccinated calves. Another new finding was that the weight gained by calves vaccinated with the trivalent MLV vaccine contrasted to the weight lost by calves sham-vaccinated with a placebo after a challenge with virulent BRSV. The presence of a “vaccine virus” is always a diagnostic dilemma, and this study provided evidence that BHV-1 in the trivalent vaccine is highly likely to be present in samples up to 9 days and likely will be negative after 14 days. BRSV is less likely than BHV-1 to be detected by PCR on nasal swabs, but BHV-1 can be present in samples up to 9 days and likely will be negative after 14 days. Results of this and previous trivalent IN MLV efficacy studies collectively support the claim that this trivalent MLV vaccine is safe and effective for intranasal vaccination of calves at one week of age or older against respiratory disease caused by BRSV, and there is a duration of immunity of at least 80 days. While the protection conferred is not complete, up to a 50% reduction in the severity of disease was noted in the vaccinated calves.

Future studies could be aimed at establishing the maximum duration of immunity and amnestic responses with additional doses of IN trivalent and or parenteral viral vaccines. It would be interesting to further characterize the elements of the mucosal response that are activated by vaccination with this IN trivalent vaccine and provide protection to enhance the health of the animal.

## Figures and Tables

**Figure 1 pathogens-13-00517-f001:**
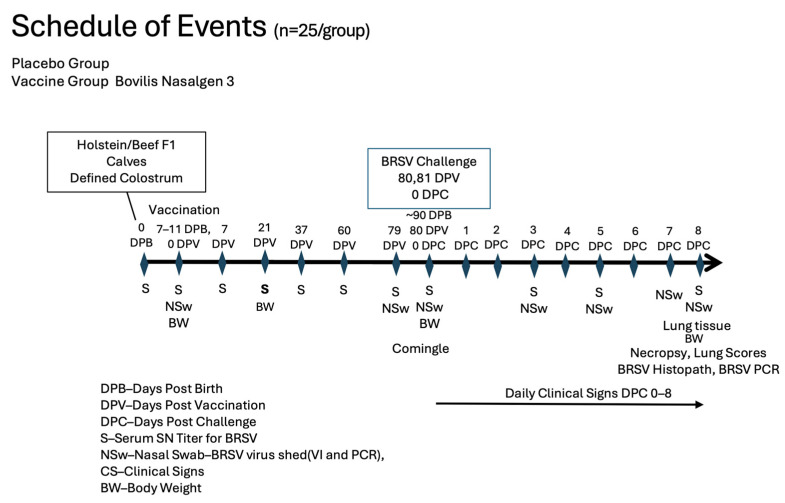
Schedule of Events. Calves were collected at birth from the dairy and fed colostrum within 24 h of birth. The calves were vaccinated at 9 ± 2 days. The animals were then sampled and challenged as indicated in the schedule of events.

**Figure 2 pathogens-13-00517-f002:**
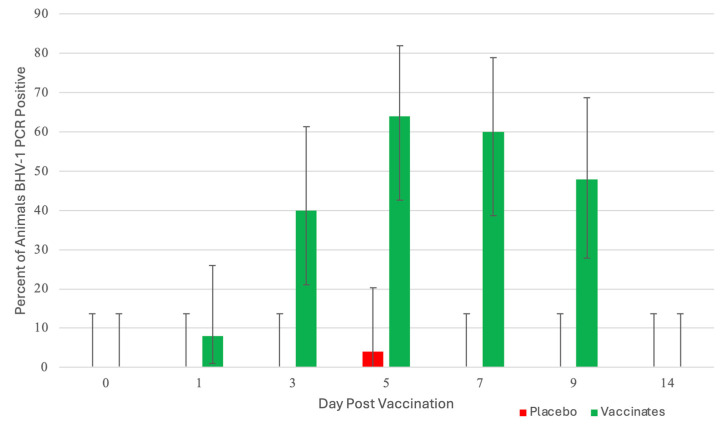
Percentage of Animals Positive for Bovine Herpesvirus Virus Nasal as detected by PCR. Nasal swabs were collected from all calves on days 0, 1, 3, 5, 7, 9, and 14 days post-vaccination (DPV) BHV-1 for detection of BHV-1 using the polymerase chain reaction (PCR) assay.

**Figure 3 pathogens-13-00517-f003:**
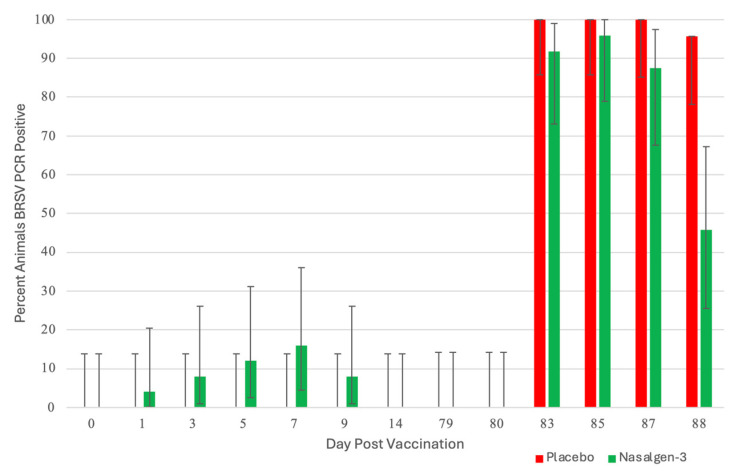
Percentage of Animals Positive for Bovine Respiratory Syncytial Virus as detected by RT–PCR. Nasal swabs were collected from all calves on days 0, 1, 3, 5, 7, 9, and 14 days post-vaccination (DPV) and on days −1, 0, 3, 5, 7, and 8 days post-challenge (DPC) for detection of BRSV using the reverse transcriptase–polymerase chain reaction (RT–PCR) assay.

**Figure 4 pathogens-13-00517-f004:**
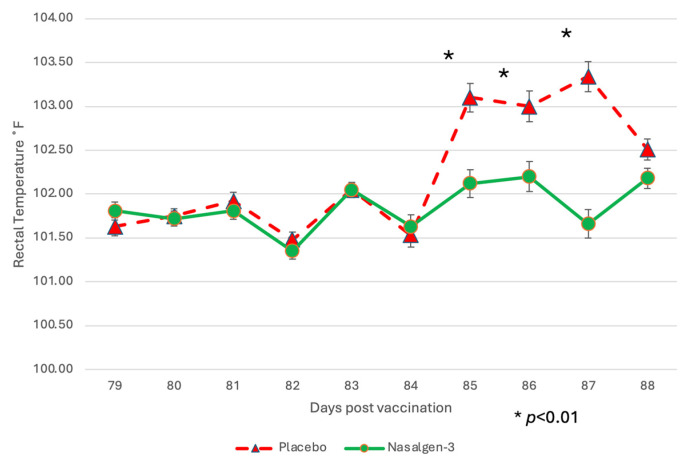
Mean Rectal Temperature. Body temperature was monitored for 2 days prior to challenge and 8 days following challenge. * *p* < 0.01.

**Figure 5 pathogens-13-00517-f005:**
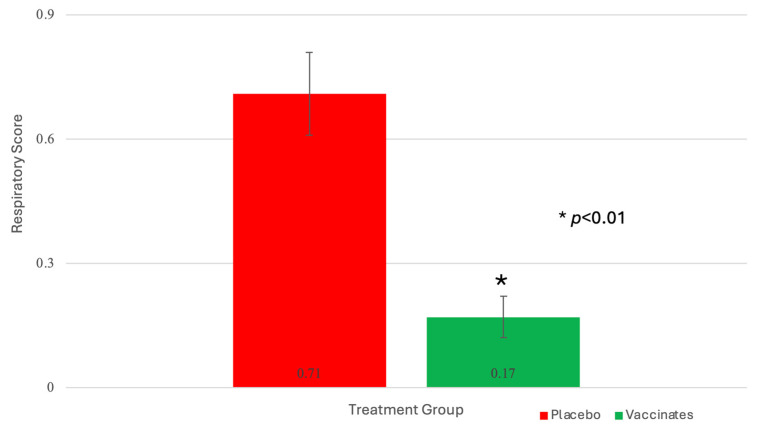
Mean Respiratory Scores. Clinical disease parameters, including general respiratory signs, were monitored for 2 days prior to challenge and 8 days following challenge. One placebo animal succumbed to the BRSV challenge. * *p* < 0.01.

**Figure 6 pathogens-13-00517-f006:**
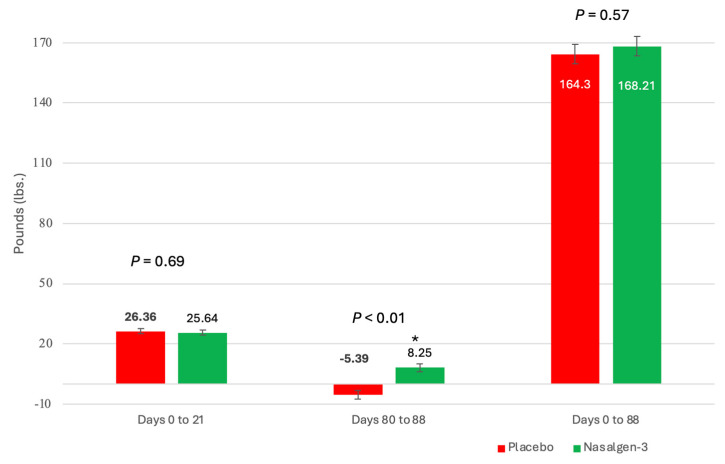
Weight Change following intranasal vaccination and BRSV challenge. Change in body weight for calves in each treatment group during the first 21 days, during the post-challenge phase, and throughout the study. * *p* < 0.01.

**Figure 7 pathogens-13-00517-f007:**
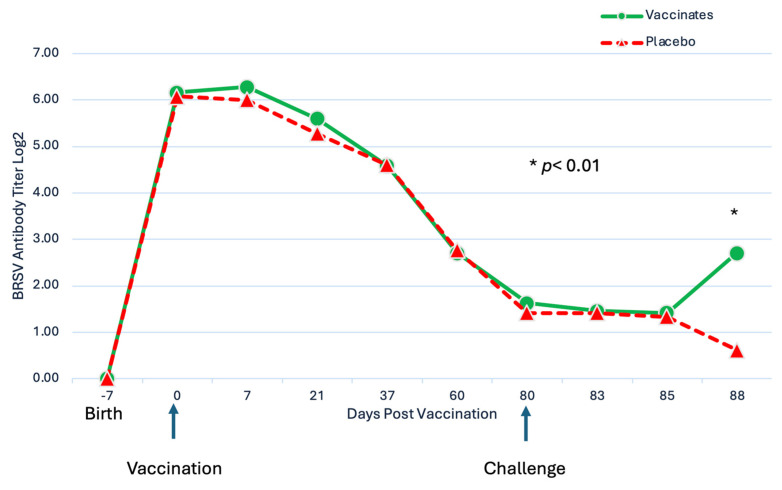
BRSV Serum Neutralization Titers. Blood samples for BRSV serum antibody analysis were obtained from the calves via jugular venipuncture at 7–21 day intervals for approximately 10 weeks prior to challenge, the day before challenge (–1 DPC), and at necropsy (8DPC). * *p* < 0.01.

**Figure 8 pathogens-13-00517-f008:**
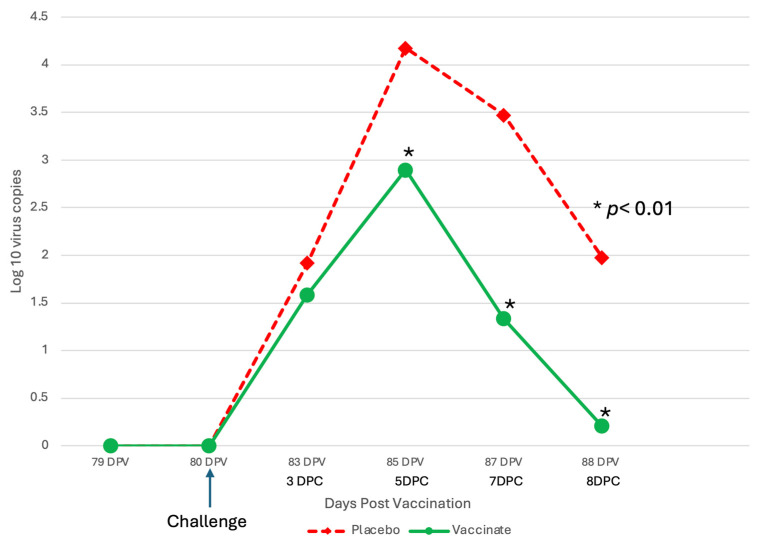
Levels of BRSV detected from nasal swabs following BRSV Challenge using RT–PCR. Nasal swabs were collected from all calves on days –1, 0, 3, 5, 7, and 8 days post-challenge (DPC) for BRSV RT–PCR assay. * *p* < 0.01.

**Figure 9 pathogens-13-00517-f009:**
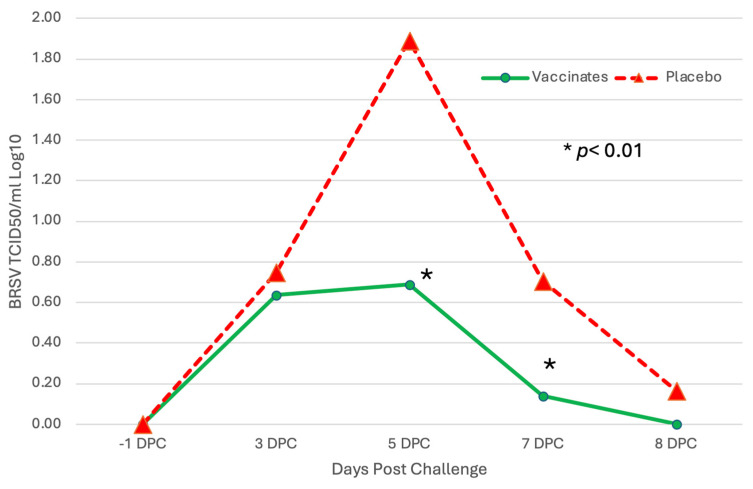
Levels of BRSV Shed as detected by virus isolation following BRSV challenge. Nasal swabs were collected from all calves on days –1, 0, 3, 5, 7, and 8 days post-challenge (DPC) for BRSV for VI assay. * *p* < 0.01.

**Figure 10 pathogens-13-00517-f010:**
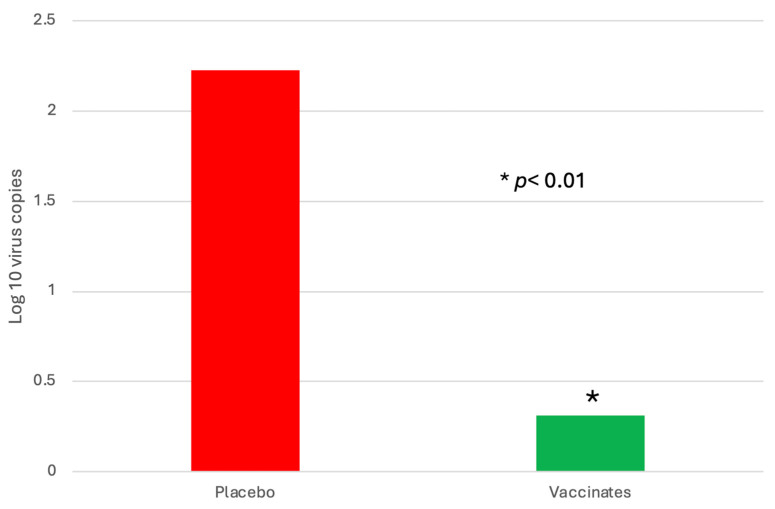
Levels of BRSV in the lung following BRSV challenge using RT–PCR. Following euthanasia, two representative areas of the lung were collected. Five grams of lung was processed with media with antibiotics, and 1 mL of supernatant was then submitted for BRSV RT–PCR. * *p* < 0.01.

**Figure 11 pathogens-13-00517-f011:**
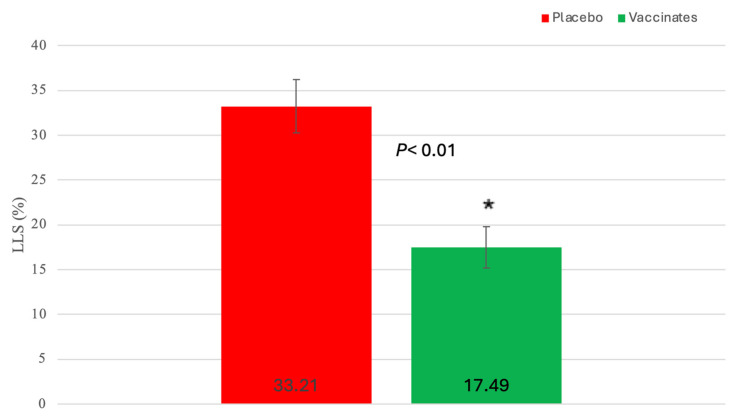
Mean lung lesion scores (LLS) on day 88 (8 days post-challenge). Lesions of the lungs were scored as a percentage of the lung involved. Each lung lobe was examined in its entirety, and the extent of lung involvement was estimated as a percentage of each lobe. * *p* < 0.01.

**Figure 12 pathogens-13-00517-f012:**
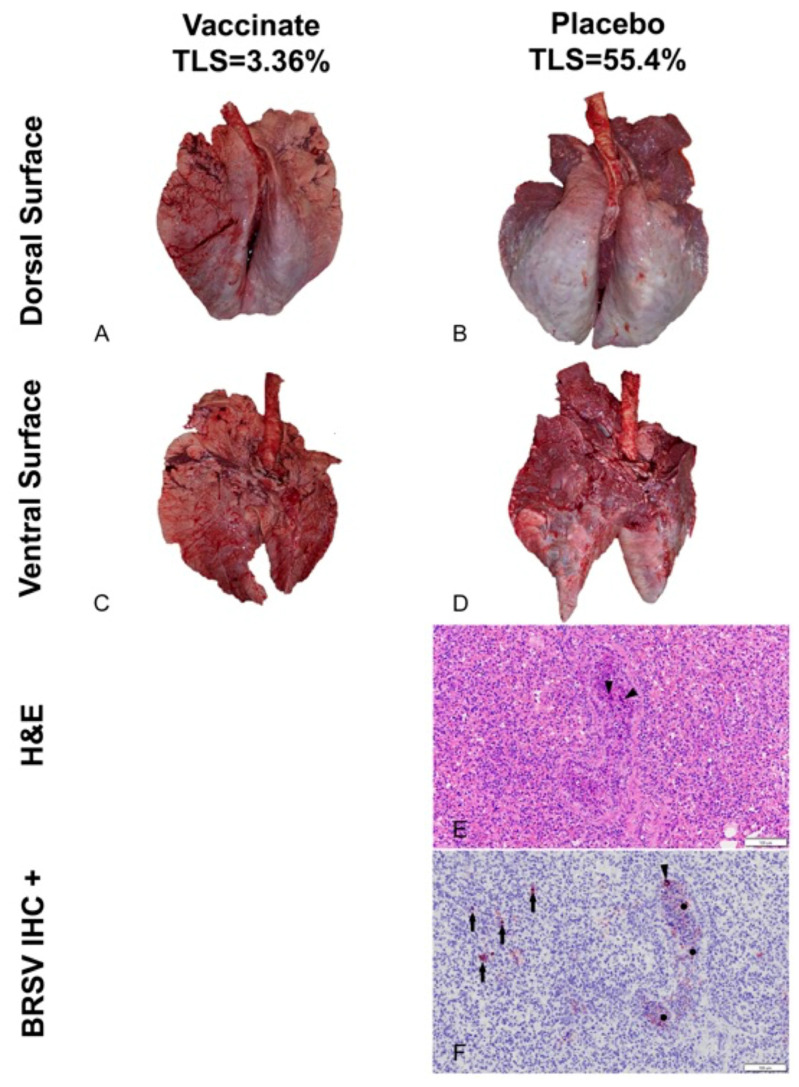
Lung Gross Lesions and Histopathology. (**A**) Vaccinate gross lesions dorsal surface; (**B**) Placebo/Control gross lesions dorsal surface; (**C**) Vaccinate gross lesions ventral surface; (**D**) Placebo/Control gross lesions ventral surface; (**E**) H & E Placebo/Control histological lesions; (**F**) Placebo/Control BRSV IHC staining.

## Data Availability

The original contributions presented in the study are included in the article, further inquiries can be directed to the corresponding author.

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
