# Peer review of "The Detection of Vaccine Virus and Protection of a Modified Live, Intranasal, Trivalent Vaccine in Neonatal, Colostrum-Fed Calves with an Experimental Bovine Respiratory Syncytial Virus Challenge"

_pathogens, 2024, doi:10.3390/pathogens13060517_

Round 1
Reviewer 1 Report
Comments and Suggestions for Authors
It would have been great if the author could add brief of signs and symptoms on the BRSV in the introduction section.
Moreover, a brief of the structure of BRSV would have been a good addition for the readers.
MM section:
Cell source
Cell conc
Control details for VI
VI and PCR reference
Incubation time for serum/virus mixture
Volume of mixture used to inoculate the cell monolayer.
No of cell passages performed while doing VI
PCR primer details and cyclic conditions
Please mention that supernatant was aliquoted for assays after centrifugation
For NS, viral dilution was made using media or PBS?
Processed NS dilutions incubation time is not mentioned after inoculating onto the monolayer. Please mention the volume of inoculation, incubation time and then volume of media added after for incubation. Please also mention that the media was not changed during the VI incubation period.
Control details should be provided here too. Positive and negative controls.
Methodology for VT should be provided in the MM section.
CO2 corrected to CO2
Reference for SOP used for flourescene microscopy
MDBK cell name and source
IBR name and reference
RTPCR reference, primers and probe details with cyclic conditions
How the animals were challenged to propagate the seed for challenge studies.
BVD RTPCR primers and cyclic conditions details. MacConkey media source and SOP reference.
IHC and histopathology details including equipment used for sectioning the tissues and H&E stain should be written in full on line 235 and not at line 242
Comma after 3 line 301
Spell check line 306
Did BHV-1 detection were also performed in the placebo group? Line 309
Shedding determined DPV and not DPC??
The sections where it states data not shown must be provided with justification of why it is not shown?
What is the protective titer of antibodies post vaccination that can withstand the challenge?
Can vaccinated and DPC virus shedding can infect non-vaccinated healthy animals in the same shed? It would have been a good study if some non-vaccinated non challenged sentinel animals should be kept with the vaccinated and challenged group to see viral transmission pattern after shedding.
Log10 should be written as Log10
Results:
Very heavy data is provided with both narrative and graphical format. It could have been much better if it can be either provided with graphs with a minimal narrative description. It is just a suggestion please.
Discussion sections: Statements line 546 needs to be reworded as this can not be justified in this study and in literature.
The statement of using single doe at lines 548 contradicts the statement at line 557 of using the INMLV. If the immunity last for 80 days than a booster dose might be required.
Viral shedding of both vaccinated and challenged group should also be discussed in detail in the discussion section.
Conclusion of the study with reference to the study hypothesis should be more concise. Suggested to please re-visit the discussion section.
Thanks
Comments on the Quality of English Language
The result data needs to be re-written in a much simpler way please.
Author Response
We appreciate the comments from the reviewers. Here are the responses to the reviewers.
Reviewer 1
It would have been great if the author could add brief of signs and symptoms on the BRSV in the introduction section.
Moreover, a brief of the structure of BRSV would have been a good addition for the readers.
Both of these suggestions were covered in lines 43-52.
MM section:
Cell source This was added in lines 163-171.
Cell conc This was added at line 169.
Control details for VI This was added 190-191 and 199-200.
VI and PCR reference All of these were already in the manuscript. References 16, 17, 21 and 22
Incubation time for serum/virus mixture This was added at line 177.
Volume of mixture used to inoculate the cell monolayer. This was added at line 178.
No of cell passages performed while doing VI. This is a single blind passage which was already in the manuscript lines 194 and 201.
PCR primer details and cyclic conditions The references include the primer and PCR conditions
Please mention that supernatant was aliquoted for assays after centrifugation. This was already in the manuscript line 186.
For NS, viral dilution was made using media or PBS? Transport media was used lines 187 and 195.
Processed NS dilutions incubation time is not mentioned after inoculating onto the monolayer. Please mention the volume of inoculation, incubation time and then volume of media added after for incubation. Please also mention that the media was not changed during the VI incubation period. Incubation time is mentioned on lines 189 and 197. Volume of the inoculation was the final volume.
Control details should be provided here too. Positive and negative controls. We referred to the same controls for PCR as the VI controls in line 207.
Methodology for VT should be provided in the MM section. I am unsure what VT refers to. We took it to refer to virus transport and we included it in line 171.
CO2 corrected to CO2 This was done
Reference for SOP used for flourescene microscopy. This was added on line 193
MDBK cell name and source This was added on lines 166-167.
IBR name and reference This was switched to BHV-1 throughout the manuscript, and the source was included in lines 169-170
RTPCR reference, primers and probe details with cyclic conditions. The reference which was already in the manuscript contains the primer, probe and assay conditions .
How the animals were challenged to propagate the seed for challenge studies. This was added on lines 213-216.
BVD RTPCR primers and cyclic conditions details. MacConkey media source and SOP reference. The reference in the manuscript contains the primer, probe and assay conditions . Added the reference and the source on lines 217-218.
IHC and histopathology details including equipment used for sectioning the tissues and H&E stain should be written in full on line 235 and not at line 242. This was written correctly but it may not have been clear that the procedures were done at the ADRDL which has been added at line 264.
Comma after 3 line 301 Done
Spell check line 306 Done
Did BHV-1 detection were also performed in the placebo group? Line 309 The BHV-1 study with the placebos was already described in lines 314-316.
Shedding determined DPV and not DPC?? This is described in lines 393-419 and was already in the manuscript.
The sections where it states data not shown must be provided with justification of why it is not shown? The BHV-1 SN serology data looks exactly like the BRSV placebo, and we didn’t do a challenge, it didn’t seem needed. The BRSV VI post challenge data is in the text, so it is “shown” but the figure is not shown (see explanation below-Very heavy data). Thanks for spotting that
What is the protective titer of antibodies post vaccination that can withstand the challenge? There are a couple of papers that show that as 1:8 can inhibit BRSV MLV vaccine responses given parenterally (see reference 12).
Can vaccinated and DPC virus shedding can infect non-vaccinated healthy animals in the same shed? It would have been a good study if some non-vaccinated non challenged sentinel animals should be kept with the vaccinated and challenged group to see viral transmission pattern after shedding. Interesting question but we didn’t do the experiment.
Log10 should be written as Log10 This was changed throughout the manuscript- it should be subscript.
Results:
Very heavy data is provided with both narrative and graphical format. It could have been much better if it can be either provided with graphs with a minimal narrative description. It is just a suggestion please. We appreciate the comment. There are three areas where it is fairly intensive and that is the PCR and VI virus shed results and the serology results. We have rearranged some of the material and moved it to the discussion (see below). Just listing percentages is not good enough because there are major shifts that involve the entire population. By listing the number of animals positive it makes the impact clearly. One of the reasons that we did not include z figure for the postchallenge animals VI results was that we already had three virus shed graphs- one that included prechallenge and post challenge that looked at % positive by PCR and then the levels of virus shed post challenge measured by both PCR and VI. The Postchallenge %positive VI data was comparable to these results and by moving to the discussion helps improve the flow.
Discussion sections: Statements line 546 needs to be reworded as this cannot be justified in this study and in literature. This has been reworded on lines 577-579.
The statement of using single doe at lines 548 contradicts the statement at line 557 of using the INMLV. If the immunity last for 80 days than a booster dose might be required. We have rewritten the preceding section so there is less ambiguity
Viral shedding of both vaccinated and challenged group should also be discussed in detail in the discussion section. We have moved material from the results into this discussion in regard to this point on lines 531-546.
Conclusion of the study with reference to the study hypothesis should be more concise. Suggested to please re-visit the discussion section. We have changed the concluding paragraph as it was redundant. Lines 594-598.
Reviewer 2 Report
Comments and Suggestions for Authors
The article provides clear and concise information about the study design, results, and conclusions. However, minor mistakes could be enhanced for better clarity.
Such as
This sentence should be corrected "administered in the face of passive immunity" as "administered in the presence of passive immunity."
This sentence should be "by intubation to fifty (50) beef dairy crossbred calves" change into "by intubation to fifty (50) beef-dairy crossbred calves."
This sentence should be corrected as "One of the most widely accepted disease protection measures is maternal antibody transference at birth “corrected as "One of the most widely accepted disease protection measures is the transference of maternal antibodies at birth."
This sentence should be changed to "Most BRD vaccine studies have focused on vaccination around weaning (just before, at or after) and then observe the effects on BRD morbidity and mortality "change to "Most BRD vaccine studies have focused on vaccination around weaning (just before, at, or after) and then observed the effects on BRD morbidity and mortality."
This article is well-written and thoroughly explained. In my opinion, no further changes are needed. After this minor revision, it can be considered acceptable.
Author Response
Reviewer 2: We appreciate the positive comments on the manuscript.
The article provides clear and concise information about the study design, results, and conclusions. However, minor mistakes could be enhanced for better clarity.
Such as
This sentence should be corrected "administered in the face of passive immunity" as "administered in the presence of passive immunity." We have made the correction.
This sentence should be "by intubation to fifty (50) beef dairy crossbred calves" change into "by intubation to fifty (50) beef-dairy crossbred calves." We have made the correction.
This sentence should be corrected as "One of the most widely accepted disease protection measures is maternal antibody transference at birth “corrected as "One of the most widely accepted disease protection measures is the transference of maternal antibodies at birth." We have made the correction.
This sentence should be changed to "Most BRD vaccine studies have focused on vaccination around weaning (just before, at or after) and then observe the effects on BRD morbidity and mortality "change to "Most BRD vaccine studies have focused on vaccination around weaning (just before, at, or after) and then observed the effects on BRD morbidity and mortality." We have made the correction.
This article is well-written and thoroughly explained. In my opinion, no further changes are needed. After this minor revision, it can be considered acceptable.
Round 2
Reviewer 1 Report
Comments and Suggestions for Authors
Please see my comments below:
Efficacy of what? Line 19
Trivalent vaccine (MLV) against which diseases and/or strains? Please clarify. Line 23
It is suggested to use the word “post” instead of “after” when referring to infection or exposure. Lines 47-48.
Please re-word lines 75-81. The flow of these sentences is difficult to understand.
Similar comments for lines 83-85 please.
The objectives in the intro should state the usage of Trivalent vaccine as mentioned in the title please. Line 88.
Please replace the word “test” to ELISA kit. Line 100.
Please use words like manufacturers recommendations instead of label directions. Line 127
Please correct sentence in lines 129-130.
Log 2 be corrected to Log2 Line 132. Please be consistent in writing it at other locations in the paper.
Lines 135-141 should be in the results section please.
The whole MM section needs to be re-written using own wordings, as the author used same sentences in almost every sub section from Reference 16 please.
Comments on the Quality of English LanguageMany sentences in the manuscript are confusing that needs to be clarified as mentioned in my comments above. The author must use own language/writing especially in the MM section please. Thanks
